# The Probiotic BB12 Induces MicroRNAs Involved in Antigen Processing and Presentation in Porcine Monocyte-Derived Dendritic Cells

**DOI:** 10.3390/ijms21030687

**Published:** 2020-01-21

**Authors:** Marlene Bravo-Parra, Marina Arenas-Padilla, Valeria Bárcenas-Preciado, Jesús Hernández, Verónica Mata-Haro

**Affiliations:** 1Laboratory of Microbiology and Immunology, Ciencia de los Alimentos, Centro de Investigación en Alimentación y Desarrollo A. C. (CIAD). Carretera Gustavo Astiazarán Rosas 46, Col. la Victoria, Hermosillo 83304, Sonora, Mexico; marlene.bravo@estudiantes.ciad.mx (M.B.-P.); maarepadi@gmail.com (M.A.-P.); valeria.bpreciado@hotmail.com (V.B.-P.); 2Laboratory of Immunology, Centro de Investigación en Alimentación y Desarrollo A. C. (CIAD), México. Carretera Gustavo Astiazarán Rosas 46, Col. la Victoria, Hermosillo 83304, Sonora, Mexico; jhdez@ciad.mx

**Keywords:** microRNAs, porcine moDCs, probiotics, MHC II pathway, IL-10, costimulatory molecules, microarray

## Abstract

MicroRNAs (miRNAs) mediate the regulation of gene expression. Several reports indicate that probiotics induce miRNA-mediated immunomodulation at different levels, such as cytokine production and the up-regulation of several markers related to antigen presentation in antigen-presenting cells. The objective of this work was to identify target genes of miRNAs that are involved in the processing and presentation of antigens in monocyte-derived dendritic cells (moDCs) stimulated with the probiotic *Bifidobacterium animalis* ssp. *lactis* BB12 (BB12). First, an in silico prediction analysis for a putative miRNA binding site within a given mRNA target was performed using RNAHybrid software with mature sequences of differentially expressed miRNAs retrieved from a Genbank data set that included BB12-stimulated and unstimulated porcine monocytes. From them, 23 genes resulted in targets of 19 miRNAs, highlighting miR-30b-3p, miR-671-5p, and miR-9858-5p, whose targets were costimulatory molecules, and were overexpressed (*p* < 0.05) in BB12-stimulated moDCs. The analysis of moDCs showed that the percentage of cells expressing SLA-DR^+^CD80^+^ decreased significantly (*p* = 0.0081) in BB12-stimulated moDCs; interleukin (IL)-10 production was unchanged at 6 h but increased after 24 h of culture in the presence of BB12 (*p* < 0.001). In summary, our results suggest that SLA-DR and CD80 can be down-regulated by miRNAs miR-30b-3p, miR-671-5p, and miR-9858-5p, while miR-671-5p targets IL-10.

## 1. Introduction

Some microorganisms of the microbiota have been used as probiotics due to their ability to inhibit pathogens, to modulate the immune response, and to improve the barrier function of the intestine [1]. One of the benefits associated with probiotic consumption is the modulation of the immune system, but the mechanisms by which probiotics modulate immune function are not completely understood [2]. Probiotics are absorbed by M cells at the intestinal level and are transported to deeper-lying lymphatic follicles where they are screened by immune-competent cells, which induce the signaling mechanisms to stimulate the release of cytokines and immunoglobulins [3]. Moreover, the possible interactions of probiotic bacteria with immune cells, such as intestinal epithelial cells (IEC), dendritic cells (DCs), and macrophages, might begin with an interaction between conserved regions present at the bacterial cell wall, also known as microbe-associated molecular patterns (MAMPs), and the pattern recognition receptors (PRR), primarily Toll-like receptors (TLRs), which are present at the immune cell’s surface [4]. The cells of the innate immune system play a crucial role in the initiation and subsequent direction of the adaptive immune responses. For instance, in vitro studies have demonstrated that lactobacilli can prime DCs to promote the development of Treg cells in a strain-specific manner [5]. Christensen et al. found that several lactobacilli strains were able to differentially affect bone-marrow-derivedDC maturation, as determined by cytokine and MHC II and Cluster of differentiation 80 (CD80) surface expression [6]. Another study showed that the commercial probiotic combination VSL#3^®^ (*Lactobacillus*, *Streptococcus*, and *Bifidobacterium* mix) increased interleukin-10 (IL-10) production, as well as the costimulatory molecules in human blood DCs. This effect was mostly due to *Bifidobacterium* species contained in the probiotic mixture [7]. We and others have demonstrated that this effect is mediated in part by the TLR2 interaction. Our research group showed, in pig monocytes stimulated with *Bifidobacterium animalis* ssp. *lactis* BB12 (BB12), the decrease in IL-10 production upon TLR2-blocking [8]. Moreover, others demonstrated that IL-10 production can be more pronounced using heat-inactivated bacteria, and this was due to IL-10 mRNA stabilization [9].

In this context, probiotics’ characteristics are exploiting to promote gastrointestinal health benefits. Nowadays, there is a growing trend to use probiotics in the diet of farm animals as an alternative to the use of antibiotics. For example, in some countries of the European Union and Japan, the use of antibiotics as growth factors has been banned in the diet of pigs to help toward looking for more natural strategies to promote and elicit the immune response of the host to the pathogens [10]. Furthermore, swine is also an excellent animal model for human studies, particularly infants, due to the great similitude in their gastrointestinal, immune, and cognitive development. Also, its omnivore feeding and a digestive tract is similar to humans. For all those reasons, it has emerged as an excellent model to study the human microbiome [11]. There is also a great similitude in the microbiota of humans and pigs, greater than other species, such as mice, despite their great use in research models [12,13].

Another variable that can modulate gene expression and in turn participate in immune modulation are microRNAs (miRNAs), which bind to the messenger RNA (mRNA) 3’UTR region of target genes, blocking its translation. As a direct consequence, miRNAs regulate many biological processes and play an important role in proliferation, differentiation, and cellular death [14,15]. It has already been demonstrated that miRNAs are involved in the regulation of the immune response toward microbiota; nevertheless, the majority of those studies have been performed in murine and human models, and even fewer of those having been performed in a porcine model [16,17]. Therefore, it is important to thoroughly understand the mechanisms of miRNAs immune regulation induced by probiotic stimuli.

In a microarray of porcine peripheral blood monocytes previously performed by our research group, cells were stimulated with *Bifidobacterium animalis* ssp. *lactis* BB12 (BB12), and upon TLR2 blocking, 52 miRNAs were differentially expressed, and a decrease in the production of IL-10 was observed. IL-10 is an anti-inflammatory cytokine, capable of regulating both innate and adaptive immunity and limiting the activation of T cells and their differentiation in lymph nodes. There are low percentages of DCs in blood circulation; thus, a frequent alternative to studying DCs’ immune biology is the in vitro generation of monocyte-derived dendritic cells (moDCs), which conserve most of the DCs qualities and characteristics [18]. DCs are distributed in most tissues, and in particular, at sites that interface with the external environment, such as the mucosa of the gastrointestinal tract, where they reside in the Peyer’s patch, lamina propria, and drain mesenteric lymph nodes.

Given that the antigen processing and presentation are natural events for DCs and other antigen-presenting cells, it is expected that some miRNAs would be expressed after the moDC stimuli with BB12 and that those miRNAs are involved in the regulation of antigen processing and presentation transcripts. We hypothesized that this effect can also be present in DCs, since IL-10 has autocrine signaling in DCs, preventing the migration toward regional lymph nodes, precluding antigenic processing and presentation, as observed in mycobacteria. In that context, IL-10 can inhibit the expression of MHC II and co-stimulatory molecules like CD80/86 [19]. In the present study, an in silico analysis was conducted to determine those miRNAs that target antigen-presenting molecules using a dataset from BB12-stimulated porcine monocytes, followed by the validation in porcine moDCs of those miRNAs that target co-stimulatory molecules with the most significant energy reactions, as well as the quantification of co-stimulatory molecules and the expression and production of IL-10.

## 2. Results

### 2.1. In Silico Prediction Analysis of the Hybridization Reaction between miRNAs and mRNAs

An in silico analysis was performed to predict a putative miRNA binding site within a given mRNA target. For this, we selected the pig data sets from the NCBI GEO dataset (GSE132995), which was originally generated by our group (Arenas-Padilla, 2018) [20] to study miRNAs involved in IL-10 regulation via TLR2 in BB12-stimulated monocytes [20]. Of the 407 miRNAs available for the pig, 67 were differentially expressed by the stimulation with BB12; of those, the 33 upregulated miRNAs with a *p*-value < 0.05 and a fluorescence signal >500 were used to perform the in silico prediction of the hybridization reaction between those miRNAs and the mRNA 3’UTR of 34 genes involved in the MHC II pathway (Appendix A). Interestingly, 19 different miRNAs target genes related to antigen presentation, such as important molecules of the MHC II pathway, cytokines, chemokines, and co-stimulatory molecules. Sixty-nine in silico hybridization reactions hada minimal free energy (mfe) ≤ −25 kcal/mol. Of the selected reactions, 11different miRNAs targeted CD74, a polypeptide also known as Class II-associated invariant chain peptide (CLIP) peptide, which participates in the formation and correct transport of MHC II protein from the endoplasmic reticulum to endocytic vesicles where antigenic peptides are loaded. CD74 binds to the cleft of MHC II protein where the antigenic peptide is loaded during the transport of the molecule to avoid MHC II binding to unrelated molecules [21]. Seven miRNAs target DNA-binding protein RFXANK (RFXANK), a subunit of a tripartite RFX complex that assembles on promoters of MHC II [22]. Likewise, TNF (tumor necrosis factor), an important member of the pro-inflammatory cytokines, was the target of six different miRNAs, and legumain (LGMN), cathepsin B precursor (CTSB), and L-selectin (SELL) genes by four miRNAs, respectively. LGMN and CTSB participate in the synthesis of the MHC II molecule. By its side, SELL is a cell surface adhesion molecule, which is necessary for dendritic cell migration during the inflammatory process [23]. Similarly, Chemokine (C-X-C motif) ligand 2 (CXCL2) and C-X-C motif chemokine 10 (CXCL10) were targeted by three and two miRNAs, respectively. These molecules are pro-inflammatory and chemokines, which can promote the recruitment of pro-inflammatory cells to the infection site, like neutrophils, basophils, and NK cells, and can induce the activation and proliferation of lymphocytes [24]. Likewise, the DNA-binding protein RFX5 (RFX5) that forms part of the complex that promotes MHC II transcription was targeted by two miRNAs, while MHC class II histocompatibility antigen SLA-DRB1 precursor (SLA-DRB1), SLA-DQ haplotype C beta chain (SLA-DQB1), SLA-DM alpha chain precursor (SLA-DMA), and SLA-DM DM beta precursor (SLA-DMB), the most common isoforms of *Sus scrofa* MHC II protein, were targeted by two miRNAs and SLA- DO beta (SLA-DOB) was targeted by just one. Interestingly, CD80 along with CD86 were targeted by the same miRNAs, miR-9858-5p and miR-9846-3p, and additionally by miR-30b-3p and miR-31, respectively. The CD80/86 is a co-stimulatory complex with a crucial role in the antigen presentation to T cells. The remaining genes were targeted by one miRNA (nuclear transcription factor Y subunit alpha (NFYA), beta (NFYB), cyclic AMP-responsive element-binding protein 1 isoform X1 (CREB1), and gamma-interferon-inducible-lysosomal thiol reductase precursor (IFI30)). The first two genes form part of a trimer composed of NFYA, NFYB, and NFYC subunits, which in the association, function as a transcription factor that joins to glutamine-rich activation domains and regulates the expression of MHC II molecules [25]. CREB1 is a transcription factor as well, and it has been implicated in the regulation of the innate immune-response-like TLR2 signaling pathway, which is activated by lipopolysaccharide and peptidoglycan stimuli [26]. IFI30 is a lysosomal reductase that is constitutively expressed in antigen-presenting cells, facilitating the generation of antigenic peptides that bind MHC II proteins toward the reduction of disulfide bonds [27].

Table 1 shows the results of the prediction in silico of miRNA–mRNA 3′UTR hybridization reactions, where miRNAs are organized from lower to higher mfe values. Nineteen miRNAs were involved with 23 antigen processing and presentation pathway molecules: miR-181b, miR-362, miR-7138-5p, miR-9802-3p, miR-9813-5p, miR-185, miR-29b, miR-30b-3p, miR-30c-1-3p, miR-31, miR-345-3p, miR-4334-5p, miR-671-5p, miR-7, miR-9816-3p, miR-9846-3p, miR-9857-5p, miR-9858-5p, and miR-99b. From there, the miRNA miR-671-5p is highlighted because it targeted nine different genes, most of which had the lowest mfe values.

### 2.2. Prediction of Differentially Expressed miRNAs Target and Gene Ontology

Forty-four upregulated miRNAs in a microarray (GEO, GSE132995) with *p* < 0.05, regardless of the fluorescence signal intensity, were also used to find targets of genes of the biological process through GO functional annotation, using the software miRpath (Figure 1). The Fisher exact test was used and a *p*-value <0.05 was set as a cut-off criterion for the GO analyses. Target genes of 11 differentially expressed miRNAs were predicted using the TarBase program. Those miRNAs were involved in the antigen presentation pathway and cytokine production (miR-671-5p, miR-30b-3p, miR-30b-5p, miR-30c-3p, miR-30e-5p, miR-296-3p, miR-345-3p) angiogenesis and cell proliferation (miR-296-3p, miR-425-5p, miR-484, let-7d-3p), and the Wnt/β-catenin signaling pathway (miR-885-5p). These pathways have an important function in the antigen processing and presentation processes.

### 2.3. moDCs Phenotype after BB12 Stimuli

The in silico prediction was made with a dataset from monocytes, and although monocytes are antigen presenting cells (APCs) in certain circumstances, the literature concurs that monocytes mainly serve as precursors for macrophages and even DCs under inflammatory conditions [28]. For that reason, the results of in silico predictions were validated in moDCs. For this, moDCs cultures were generated in vitro and challenged with BB12. Figure 2a shows the selection of the cell population based on forward and side scatters, and Figure 2b shows the singlet cell selection [29,30]. Figure 2c shows a representative dot-plot of unstimulated moDCs (top) and those stimulated with BB12 (bottom). The population of interest is cells SLA-DR^+^/CD80^+^, which represent a characteristic phenotype of moDCs [31]. A decrease in the percentages of moDCs expressing the phenotype SLA-DR^+^-CD80^+^ moDCs (** *p* = 0.0081) was observed in cells stimulated with a BB12 probiotic (Figure 2d).

### 2.4. Relative Expression of IL-10 and miRNAs

Previous data from our research group show that BB12 increases the IL-10 expression in monocytes after 4 h [8]. To determine the expression of IL-10 mRNA, moDCs were cultured in the presence of BB12 over6 h, and as a control, moDCs were cultured in the presence of media alone (Figure 3a). No significant differences were found in the relative expression of IL-10 using RT-qPCR in BB12-stimulated moDCs with respect to unstimulated cells (*p* > 0.05). However, the production of IL-10 was increased in moDCs stimulated with BB12 (Figure 3b) after 24 h in culture, in agreement with the results from the monocytes. Conversely, the production of IL-12 had no statistical difference in both groups (Figure 3c).

Furthermore, three of the miRNAs (miR-30b-3p, miR-671-5p, and miR-9858-5p) that targeted clue genes in the antigen presentation process that were found in the in silico analysis were chosen to validate their expression in the BBL12-treatedmoDCs. Those miRNAs were chosen due to the high number of target genes (3, 5, and 9, respectively) in the prediction analysis. The relative expression of miRNAs was performed using RT-qPCR (Figure 4). The results showed that the three miRNAs had a significant increase in BB12-stimulated moDCs with respect to unstimulated control cells. 

## 3. Discussion

Several reports have focused on the capacity of probiotics to modulate the immune response in different steps. The recent studies on the regulation of signaling molecules by miRNAs led us to investigate whether miRNAs could be involved in the probiotic effect. For this, we used moDCs, an important link between innate and adaptive responses. Using data previously published in the GEO database, we first selected a dataset that included monocytes stimulated with BB12 to determine a change in the expression of miRNAs upon stimulation with BB12. For that, we found 112miRNAs that changed their expression. We then selected 33 up-regulated miRNAs to perform the in silico analysis to predict the putative mRNA binding targets. We analyzed the mfe of the predicted hybridization reactions and found that miRNA miR-671-5p targeted nine different miRNAs with the best scores; a lower value has the best prediction for a target mRNA [32]. Furthermore, the reaction with the lower score was with IL-10 mRNA. Interestingly, the miRNA miR-671-5p hybridized with IL-10 mRNA 3′UTR. This may involve miR-671-5p down-regulating the IL-10 translation since it was overexpressed in the microarray performed in monocytes. However, in the present study, no significant changes in the IL-10 relative expression were found between unstimulated and BB12-stimulated moDCs at 6 h, which could be attributed to the low IL-10 production capacity of moDCs. Given that IL-10 has an autocrine regulation, we decided to quantify the IL-10 production using Enzyme-Linked ImmunoSorbent Assay (ELISA); after 24 h of stimuli with BB12, an increase of IL-10 protein was observed. Likewise, the production of IL-12 was unchanged using ELISA under the same conditions in moDCs. The 3′UTR sequence of IL-12 mRNA was included in the in silico prediction analysis, but interestingly, it was not targeted by any of the miRNAs included in the microarray. BB12 promotes anti-inflammatory profile in monocytes and moDCs, as observed in this work.

The in silico prediction analysis showed that miR-671-5p was significantly joined to other genes like CD74, RFXANK, SLA-DQB1, NFYA, NFYC, LGMN, TNF, and SELL, indicating that they could be involved in a gene regulatory network of antigen processing and presentation. Also, miR-671-5p, miR-30b-3p, and miR-9858-5p were up-regulated in BB12-moDCs (6 h) with respect to unstimulated cells (*p* < 0.05) [33]. All miRNAs work through translation suppression mechanisms, but their functions in signaling networks need not be simply repressive. They have a dual function (up-regulation and down-regulation) in the same pathway depending on the stimuli, participating in feedback and feed-forward loops, which could explain in part, the up-regulation of 671-5p in monocytes and moDCs, the overexpression of IL-10 in monocytes, but no changes in IL-10 expression in moDCs [34]. 

At the time of this study, no published information about the porcine target of miRNA miR-671-5p had been found. Although, due to the increase of investigations of the neuro-gastrointestinal axis in recent years, miR-671-5p has been reported as being down-regulated in epilepsy in humans and mouse, and results are interesting because in our research group, this miRNA was up-regulated in response to a probiotic, and these microorganisms are known to provide beneficial effects at local and systemic levels. The up-regulation of miR-671-5p and miR-30b-3p was also observed in human peripheral blood mononuclear cells (PBMCs) stimulated with heat-treated *Lactobacillus paracasei* NCC 2461 [9]. Another study found miR-671-5p expression was significantly decreased in breast cancer via down-regulation of forkhead box protein M1 FOXM1 expression, an oncogene transcription factor. Also, the overexpression of miR-671-5p attenuated the proliferation and invasion in breast cancer cell lines [35].

Since there is no available program to perform the prediction of target genes of porcine miRNAs, results of GO analysis include just those miRNAs and target genes reported in humans. There were 79 biological processes related to 11 miRNAs. A variety of biological processes were found, such as those implicated with metabolism, cellular processes, and mRNA regulation process, where miRNAs may participate, providing signaling pathways and an innate immune response. The miRNAs miR-30b-3p and miR-671-5p were implicated in these processes, but of note, antigen processing and presentation process did not appear in the heatmap. However, activation of signaling pathways and innate immune responses are processes that happen necessarily before the antigen process and presentation; therefore, those processes could be influencing the antigen process and presentation. Nevertheless, miRNA sequences are very similar between humans and swine, but the sequences of targets can vary.

On the other hand, the percentage of SLA-DR^+^CD-80^+^ decreased significantly after 6 h of stimuli with BB12 (Figure 4) compared to unstimulated moDCs. These findings imply that BB12-stimulated cells could be undergoing an immunomodulatory process that involves the down-regulation of these co-stimulatory molecules, and as previously hypothesized, this down-regulation could be mediated by miRNAs. Many co-stimulatory molecules are modulated during the cellular activation process. Some of those may be suffering a post-transcriptional regulation. In our study, the number of SLA-DR^+^CD-80^+^-expressing BB12-stimulatedmoDCs decreased after 6 h compared with the unstimulated moDCs. Thus, in BB12-stimulated moDCs, the decreased expression of those costimulatory proteins could be mediated by the up-regulation of miRNAs. In the in silico analysis, miR-345-3p targeted the SLA-DRB1 molecule (mfe = −37.8 kcal/mol), miR-30b-3p hybridized with CD80 (mfe = −28.8), and miR-9858-5p hybridized with CD80 and CD86 (mfe = −27 and −28.5, respectively). Using RT-qPCR, we found an increase in the relative expression of miR-30b-3p and miR-9858-5p; therefore, these miRNAs could be participating in the modulation of SLA-DR and CD80/86 molecules by blocking its translation. However, we thought that at the time of our experiment, moDCs were interacting with the bacteria, inducing the expression of a variety of miRNAs, which could be due to the immunomodulatory process. MiRNAs are some of the molecules that can mediate post-transcriptional modifications, which means that miRNAs can be expressed together with other transcripts (mRNAs) that codify proteins involved in immune response, for example, to participate in its regulation. In an experiment performed in humans, miR-30b was reportedly involved in attenuating the uptake and processing of the soluble antigen ovalbumin in primary macrophages and dendritic cells. This result is interesting because antigen phagocytosis and the later processing are events that must precede the antigen presentation process. Furthermore, it could start as early as approximately 6h after antigen recognition [36].

Although the inhibition in the expression of molecules involved in antigen processing and presentation process seems to be risky, like in evasion strategies of immune response by pathogens, BB12 is a probiotic bacterium that has the status of GRAS (generally recognized as safe) [37]. In that sense, the behavior observed in this work could be part of the cross-talking of symbiotic microorganisms and its host.

Nevertheless, the majority of in vitro experiments with moDCs employed an adjuvant to boost the maturation/activation of these cells. This must also take into account the fact that the moDCs in our study could be in the immature stage, which needs to be confirmed using a greater battery of cellular markers than those reported herein. Likewise, the silencing of miR-30b-3p, miR-671-5p, and miR-9858-5p through oligonucleotide transfection of moDC stimulated with BB12, and later quantification of antigen processing and presentation molecules and production of cytokines, would help to clarify the participation of these miRNAs in the regulation of these molecules. 

## 4. Materials and Methods

### 4.1. In Silico Prediction of the miRNA–mRNA 3′UTR Hybridization Reaction

The bio-informatic tool Kyoto Encyclopedia of Genes and Genomes (KEGG) (https://www.genome.jp/kegg-bin/show_pathway?ssc04612) was used to find genes involved with antigen processing and presentation pathway via MHC II from exogenous antigens. Additionally, cytokines, chemokines, and co-stimulatory molecules were included [19]. The mature sequences of differentially expressed (up-regulated) miRNAs from control and BB12-stimulated monocytes were collected from microarrays deposited in GEO (GSE132995) https://www.ncbi.nlm.nih.gov/geo/query/acc.cgi?acc=GPL26801. The porcine (*Susscrofa*) mRNA 3′UTR sequences were obtained from the Ensembl database using the Biomart toolkit (http://www.biomart.org/). The in silico prediction of miRNA-mRNA 3′UTR hybridization reaction was performed through RNAhybrid (https://bibiserv.cebitec.uni-bielefeld.de/rnahybrid?id=rnahybrid_view_submission) under the following conditions: helix constraint from two to seven nucleotides, one hit per target, maximal number of unpaired nucleotide equal to four, the minimal free energy (mfe) of the hybridization reaction was lower than−25 kcal/mol and no G:U in the seed region.

MiRpath software (http://snf-515788.vm.okeanos.grnet.gr/) was employed to perform Enriched Gene Ontology (GO) analysis to find target genes of differentially expressed miRNAs that were involved in the biological process. Since there is no specific program to predict target genes of swine, we used the information of human miRNAs and gene targets from TarBase v7.0 and Targetscan software.

### 4.2. Porcine Monocytes Isolation from Peripheral Blood

Peripheral blood was obtained from the anterior vena cava of healthy pigs from 4–6 weeks of age from the experimental farm of Centro de Investigación en Alimentación y Desarrollo, A.C., (CIAD). The blood was collected in tubes with heparin (10 IU/mL), and were then carefully deposited on a Ficoll-Hypaque (GE Healthcare Bio-Sciences, Uppsala, Sweden) layer, in a 3:1 portion of blood:Ficoll. Next, tubes were centrifuged for 35 min in a refrigerated centrifuge at 1600 rpm. Mononuclear cells were isolated from the interface and washed with a 4-°C phosphate buffer saline (PBS; 137 mM NaCl, 2.7 mM KCl, 10 mM Na_2_HPO_4_, and 4.2 mM KH_2_PO_4_). Monocytes were isolated using plastic adherence from mononuclear cells cultured overnight in six-well plates. 

### 4.3. Monocyte Dendritic Cell (moDC) Generation

After isolation, cells were resuspended in Dulbecco’s Modified Eagle’s Medium (DMEM, Invitrogen Co., Carlsbad, CA, USA) supplemented with L-glutamine (10µL/mL), gentamicin (50 µg/mL), penicillin/streptomycin (100 UI/mL), amphotericin B (2.5 µg/mL), β-mercaptoethanol (0.033%), and 10% fetal bovine serum (FBS). All reagents from Sigma-Aldrich (Sigma-Aldrich, St. Louis, MO, USA) unless otherwise stated. The medium was supplemented with recombinant porcine GM-CSF (20 ng/mL) and IL-4 (25 ng/mL) (R&D Systems, Abingdon, UK) to achieve moDCs differentiation [38]. Cell cultures were performed in 12-well plates (Corning Inc., Corning, NY, USA) and incubated for five days at 37 °C and 5% CO_2_ (with the addition of cytokines at day three).

### 4.4. Stimulation of moDCs with BB12

*Bifidobacterium animalis* ssp. *lactis* BB12, a commonly used probiotic, was cultivated on Man Rogosa Sharpe broth (MRS; BD Difco, Sparks, MD, USA) supplemented with 0.05% of cysteine (Sigma-Aldrich, St. Louis, MO, USA) at 37 °C for six days and 24 h in anaerobiosis using effervescent tablets for CO_2_ production [39,40]. The bacterial culture was centrifuged and the precipitate was resuspended in PBS. The concentration of bacterial suspension was determined using an optical density (600nm) and adjusted according to the equation y = (7 × 10^−9^)x + 0.003. The moDCs were cultured in 12-well plates (5 × 10^5^cells/mL in 1000 μL volume) and incubated with BB12 at 1:100 cell:bacteria ratio for 6 h for RT-qPCR and flow cytometry or 24 h for ELISA. Unstimulated cells were used as a negative control. 

### 4.5. Flow Cytometry Analysis

The expression of co-stimulatory molecules and antigen-presentation molecules were analyzed using flow cytometry. Both unstimulated cells and cells stimulated with BB12 were centrifuged, the supernatant was discarded, and the cells were resuspended in a FACS Buffer (PBS 1X, 1% BSA, and 0.1% NaN_3_) in a ratio of 1:20. Fluorochrome-conjugated antibodies anti-SLA-DR (BD Biosciences, San Jose, CA, USA) and anti-CD80 (BD Biosciences, San Jose, CA, USA) were treated with Alexa fluor-488 and APC-H7, respectively, and incubated for 30 min in the dark at 4 °C. Then, cells were washed in FACS buffer and analyzed using flow cytometry with a FACSCanto II (Becton Dickinson, San José, CA, USA).

### 4.6. Relative Quantification of Cytokines Expression UsingRT-qPCR

The total RNA from the moDCs was extracted using TRIsure (Sigma-Aldrich, St. Louis, MO, USA), according to the manufacturer’s instructions. The purity and concentration of RNA were determined using NanoDrop ND-1000 (Thermo Fisher Scientific, Wilmington, DE, USA). Total RNA was stored at −20 °C until further use. Total RNA from moDCs were treated with Turbo DNase I (Invitrogen by Thermo Fisher Scientific, Vilnius, Lithuania) to eliminate genomic DNA from the RNA samples following this protocol in the thermocycler: 30 min at 37 °C, 10 min at 65 °C, and then a cooling step at 10 °C. After the DNase treatment, RNA was used to perform quantitative reverse transcriptase PCR (RT-qPCR) reactions on a QuantStudio 6 Flex thermocycler (Applied Biosystems, Carlsbad, CA, USA). The following hydrolysis probe sequences and primers were employed to quantify the IL-10 transcripts’ expression: forward (5′–3′) TGAGAACAGCTGCATCCACTTC, reverse (5′–3′) TCTGGTCCTTCGTTTGAAAGAAA, probe: 6-FAM^TM^-CAACCAGCCTGCCCCACATGC-MGB Eclipse^®^. Also, the transcripts expression of peptidylprolyl isomerase A (PPIA, cyclophilin A) were quantified as constitutive gene PPIA: forward (5′–3′) GCCATGGAGCGCTTTGG, reverse (5′–3′) TTATTAGATTTGTCCACAGTCAGCAAT, Probe (5′–3′) 6-FAM^TM^-TGATCTTCTTGCTGGTCTTGCCATTCCT-MGB Eclipse^®^. The enzyme SuperScript III Platinum One-Step qRT-PCR Kit (Invitrogen^TM^) was used to perform reactions in a reaction mix of 6 mM MgSO_4_ and 50 ng RNA in a final volume of 25 µL. The reactions were performed under the following conditions: reverse transcription (RT) at 50 °C for 15 min, 95 °C for 2 min, qPCR step (40 cycles), 95 °C for 15 s, 56 °C for 30 s, and 72 °C for 30 s. The relative transcript levels were determined using the 2^−ΔΔ*C*T^ method, and PPIA relative quantification was used for normalization.

### 4.7. Relative Quantification of miRNA Expression by RT-qPCR

Those miRNAs with significant results in the in silico prediction analysis were confirmed using RT-qPCR in a 6 h culture of BB12-stimulated and unstimulated moDCs. Total RNA from moDCs were used to perform the reverse transcription (RT) through the preparation of mixed RT universal, 10X PAP buffer, 1mMATP, 10 µM RT-primer, 1 mM dNTPs mix, RT MU-LV (NEB), and Poly (A) polymerase (NEB) per reaction. cDNAs synthesized previously were used to perform qPCR with the enzyme 5x HOT FIREpol^®^ EvaGreen^®^ ROX (Solis Biodyne, Tartu, Estonia). The qPCR reactions were performed on a QuantStudio 6 Flex thermocycler (Applied Biosystems, Carlsbad, CA, USA).

### 4.8. MicroRNAs Primer Sequences

RT-qPCR was performed according to the protocol published by Mentzel and Skovgaard [41]. For cDNA synthesis, we used M-MuLV reverse transcriptase (New England Biolabs, Ipswich, MA, USA), and the conditions for reverse transcription reaction were 42 °C for 60 min, 95 °Cfor 5 min, followed by qPCR (50 °C for 2 min), and 95 °C, 1 cycle; and 95 °C 15 s and 60 °C 1 min, 40 cycles. Real-time PCR reactions were performed in duplicate on a QuantStudio 6.0 Flex real-time PCR system (Thermo Fisher Scientific) in a 10 μL final volume, including 2 μL HOT FIREPol^®^ EvaGreen^®^ qPCR Mix Plus (Solis Biodyne), 1 μL of each primer at 10 μM, 1 μL of a 1:4 dilution of the cDNA, and 5 μL of PCR water. The miRNAs primers (Table 2) were designed using the publicly available software miRprimerdesing3 [42]. The relative miRNAs’ expression level was calculated using the 2^−∆∆*C*T^ method and log2 was used for expressing the fold change.

### 4.9. Quantification of Cytokines UsingELISA

The moDCs were stimulated for 24 h with BB12. After that time, supernatants were collected and centrifuged to remove cells. The production of IL-10 and IL-12 cytokines was quantified using ELISA according to the manufacturer’s instructions (R&D Systems, Minneapolis, MN, USA).

### 4.10. Statistical Analysis

Statistical analyses were performed using GraphPad Prism, version 5.0 (GraphPad Software, Inc). The experiments were performed independently. The statistical tests used for parametrical data analysis were one-way analysis of variance (ANOVA) and Student’s *t*-tests. For no-parametrical data, Mann–Whitney’s test was used.

## 5. Conclusions

The IL-10 is an anti-inflammatory cytokine capable of stopping the antigen processing and presentation. In this study, IL-10 was produced from moDCs usingBB12 stimuli. Since IL-10 can prevent antigen processing and the presentation process, an in silico prediction was performed to identify miRNAs that participated in the MHC II pathway through analysis of a hybridization reaction between 3′UTR mRNAs and mature miRNA sequences, as expressed by moDCs with and without BB12 stimuli, plus validation of their relative expression. The results suggested that the probiotic BB12 induced the expression of miRNAs capable of targeting genes involved in the formation and transport of the SLA-DR, and cytokines and chemokines participating in this process. Furthermore, the miRNAs miR-30b-3p, miR-671-5p, and miR-9858 were upregulated in BB12-stimulated moDCs and could be participating in the modulation of SLA-DR and CD80/86 molecules’ expression. We described for the first time the potential of these miRNAs to participate in the regulation of the MHC II pathway in a porcine model in response to probiotic stimuli. However, more experiments need to be done to further investigate this.

## Figures and Tables

**Figure 1 ijms-21-00687-f001:**
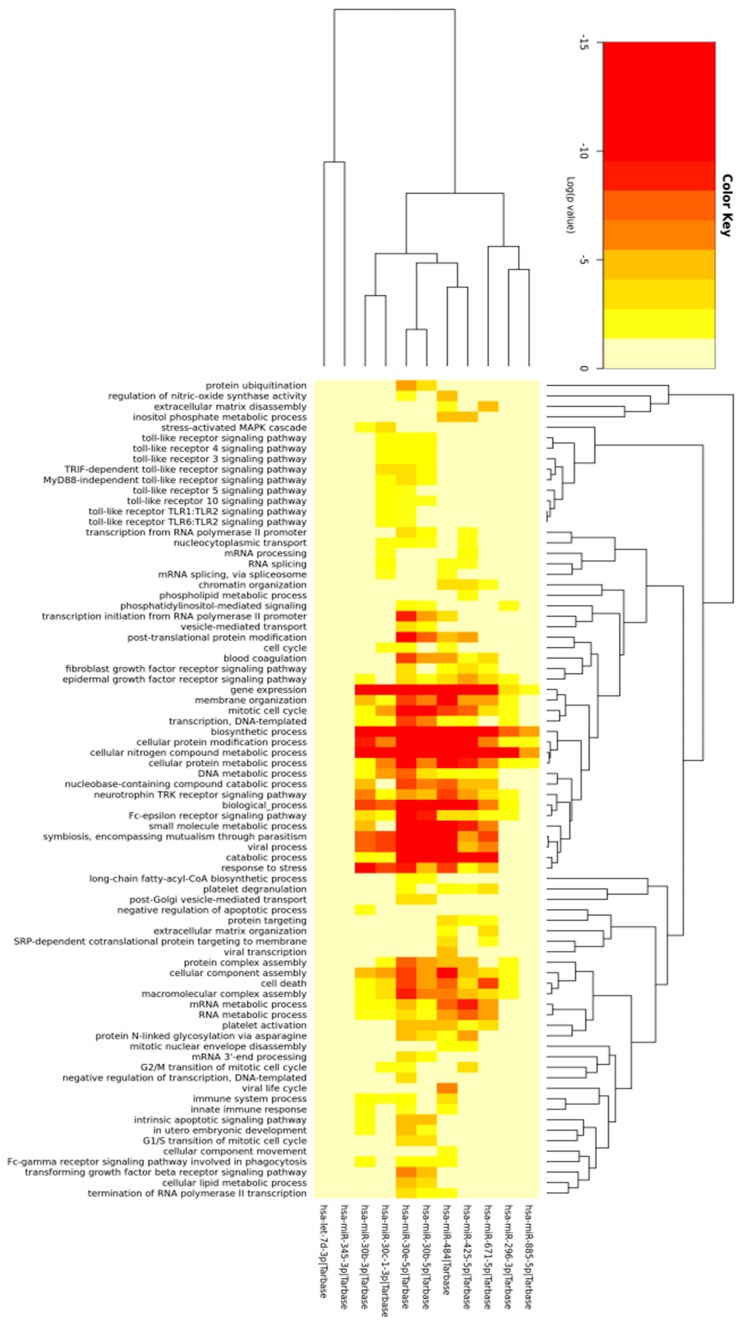
Heatmap from the Enriched Gene Ontology (GO) analysis. Eleven miRNAs from BB12-stimulated and unstimulated monocytes that were differentially expressed (GEO, GSE132995) resulted in the target of genes involved in 79 biological process in the GO analysis. The Fisher exact test (with hypergeometric distribution) and a *p*-value < 0.05 were used.

**Figure 2 ijms-21-00687-f002:**
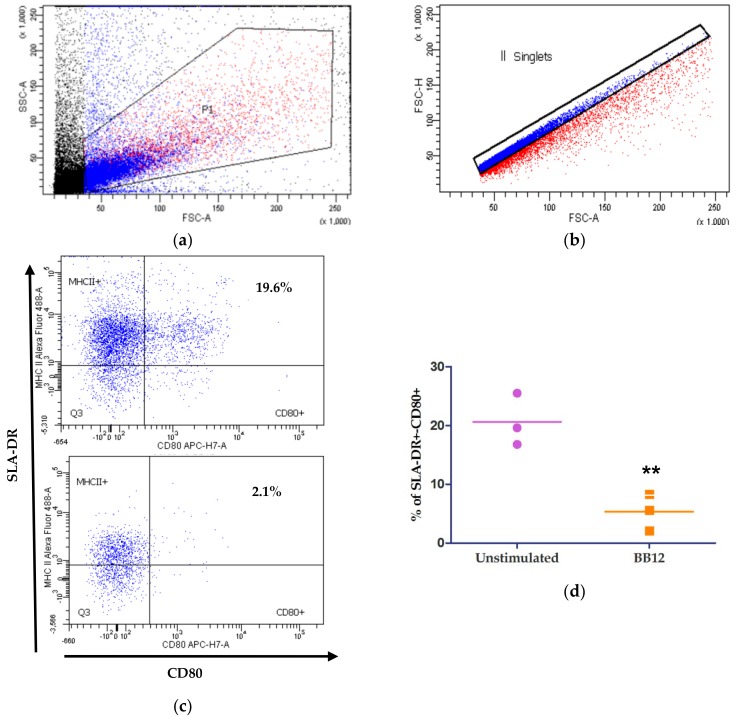
Percentage of SLA-DR^+^-CD80^+^ BB12-stimulated (6 h) moDCs decreased with respect to unstimulated moDCs. The phenotype of unstimulated and BB12-stimulated moDCs. Monocyte-derived DCs unstimulated (top) or stimulated with BB12 (bottom) for 6 h, stained for SLA-DR and CD80, and analyzed using flow cytometry. The cell population was first selected according to their size and complexity (**a**), then singlets were selected (**b**) and SLA-DR and CD80 were evaluated (**c**) (top graph features unstimulated moDCs and the bottom graph features BB12-stimulatedmoDCs; the dot plot is representative of *n* = 3). (**d**) The percentage of SLA-DR^+^-CD80^+^BB12-stimulated moDCs (BB12) decreased with respect to the unstimulated (control) moDCs (********
*p =* 0.0081). Data were analyzed using a *t*-test, *n* = 3.

**Figure 3 ijms-21-00687-f003:**
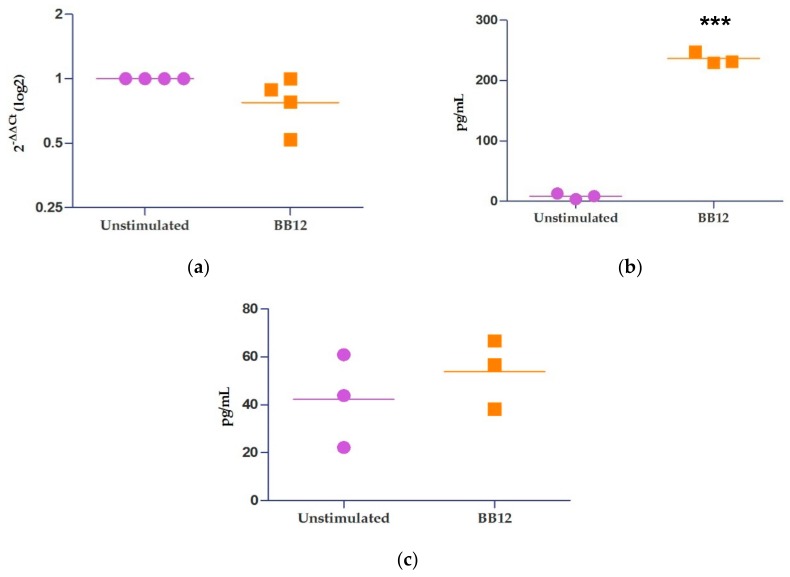
Production of IL-10, but not its expression is increased in BB12-stimulated moDCs. MoDCs were either left unstimulated or stimulated with BB12 (BB12) for 6 h; then, RNA was extracted from the cells and the relative expression of IL-10 was determined using RT-qPCR via the 2^−ΔΔ*C*t^ method. Peptidylprolyl Isomerase A (PPIA) expression was used as a housekeeping gene to normalize the data (**a**), y-axis represents foldchange of IL-10 cytokine from unstimulated or BB12-stimulated moDCs. Culture supernatants of moDCs that were unstimulated or stimulated with BB12 for 24 h were assayed for (**b**) IL-10 and (**c**) IL-12 using Enzyme-Linked ImmunoSorbent Assay (ELISA), where y-axis represents picograms per milliliter. Data were analyzed using a *t*-test, *n* = 4, *** *p*<0.001. Graphs represent the geometric mean (**a**) and mean (**b**, **c**). Each point represents one pig and an independent experiment.

**Figure 4 ijms-21-00687-f004:**
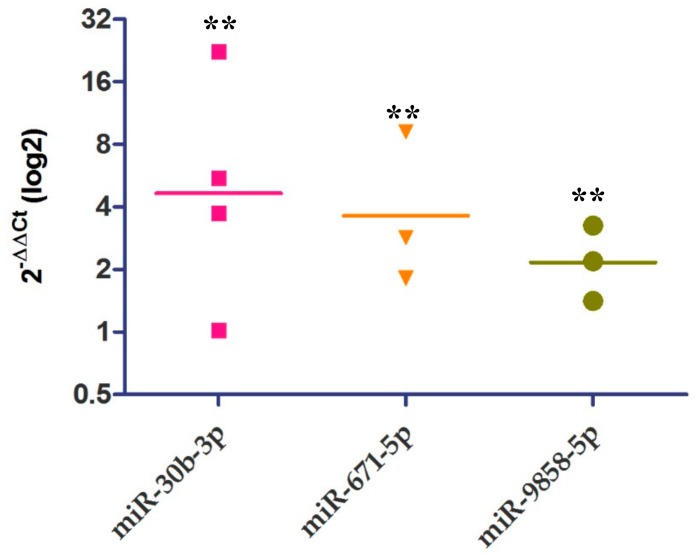
MiRNAs are increased after BB12 stimulation. The relative expression of miRNAs miR-30b-3p, miR-671-5p, and miR-9858-5p was determined using RT-qPCR in unstimulated and BB12-stimulated (BB12) moDCs via the 2^−ΔΔ*C*t^ method. The graph shows the geometric mean values of at least three independent experiments (each point represents one pig); comparisons were made between unstimulated versus BB12-stimulated moDCs for each miRNA (*p* < 0.05**), and y-axis represents foldchange relative expression. Mann–Whitney test for non-parametric data was used. The expression of ssc-U6 was used to normalize the data.

**Table 1 ijms-21-00687-t001:** In silico prediction of miRNA-mRNA 3’UTR hybridization reactions. mfe: Minimal Free Energy.

miRNA	Target	Length	Position	Mfe(kcal/mol)
*ssc-miR-671-5p*	*IL10*	735	*435*	−41.2
*ssc-miR-671-5p*	*CD74*	2784	*1507*	−38.0
*ssc-miR-671-5p*	*RFXANK*	1758	*163*	−38.0
*ssc-miR-345-3p*	*SLA-DRB1*	341	*111*	−37.8
*ssc-miR-671-5p*	*SLA-DQB1*	357	*121*	−32.0
*ssc-miR-671-5p*	*NFYA*	2590	*1722*	−31.8
*ssc-miR-671-5p*	*LGMN*	1544	720	−31.4
*ssc-miR-671-5p*	*NFYC*	743	*590*	−31.3
*ssc-miR-4334-5p*	*RFXANK*	1758	*828*	−31.0
*ssc-miR-4334-5p*	*CTSB*	948	*876*	−30.7
*ssc-miR-7138-5p*	*NFYB*	1802	*1342*	−29.9
*ssc-miR-9816-3p*	*RFXANK*	1758	*693*	−29.5
*ssc-miR-671-5p*	*TNF*	1926	197	−29.4
*ssc-miR-185*	*IL1B2*	583	*170*	−29.4
*ssc-miR-4334-5p*	*CD74*	2784	1507	−29.3
*ssc-miR-4334-5p*	*SLA-DMA*	225	131	−29.0
ssc-miR-30b-3p	*CD80*	1492	*920*	−28.8
*ssc-miR-31*	*CXCL10*	3643	*1848*	−28.8
*ssc-miR-9802-3p*	*RFX5*	1521	*139*	−28.7
*ssc-miR-9858-5p*	*TNF*	1926	*916*	−28.7
*ssc-miR-30c-1-3p*	*CD74*	2784	*1084*	−28.5
*ssc-miR-29b*	*IFI30*	228	*164*	−28.5
*ssc-miR-9858-5p*	*CD86*	3967	*252*	−28.5
ssc-miR-30b-3p	*SELL*	*2974*	509	−28.1
*ssc-miR-30c-1-3p*	*SELL*	2974	*2652*	−28.0
*ssc-miR-9857-5p*	*NFYC*	743	16	−28.0
*ssc-miR-31*	*CD74*	2784	27	−27.6
*ssc-miR-31*	*RFX5*	1521	117	−27.6
*ssc-miR-185*	*SELL*	2974	*1301*	−27.5
*ssc-miR-185*	*RFXANK*	1758	*1591*	−27.4
*ssc-miR-7138-5p*	*CD74*	2784	*2265*	−27.4
*ssc-miR-31*	*SLA-DQB1*	357	40	−27.3
*ssc-miR-185*	*IL10*	735	120	−27.1
*ssc-miR-9816-3p*	*TNF*	1926	*1745*	−27.1
ssc-miR-30b-3p	*CD74*	2784	1084	−27.0
*ssc-miR-9858-5p*	*CD80*	1492	*928*	−27.0
*ssc-miR-362*	*CD74*	2784	*124*	−26.8
*ssc-miR-7138-5p*	*TNF*	1926	*203*	−26.6
*ssc-miR-4334-5p*	*CXCL2*	712	*400*	−26.5
*ssc-miR-185*	*SLA-DMA*	225	*64*	−26.4
*ssc-miR-9846-3p*	*CXCL2*	712	*401*	−26.4
*ssc-miR-31*	*NFYC*	743	*479*	−26.4
*ssc-miR-9857-5p*	*SLA-DOB*	1030	*254*	−26.4
*ssc-miR-362*	*TNF*	1926	*1624*	−26.3
*ssc-miR-9813-5p*	*RFXANK*	1758	*688*	−26.1
*ssc-miR-9858-5p*	*SLA-DOB*	1030	*532*	−26.0
*ssc-miR-9816-3p*	*CXCL10*	3643	*876*	−26.0
*ssc-miR-9858-5p*	*CXCL10*	3643	*2094*	−25.9
*ssc-miR-185*	*CTSB*	948	*829*	−25.9
*ssc-miR-185*	*CD74*	2784	1086	−25.9
*ssc-miR-9846-3p*	*CD86*	3967	*2423*	−25.9
*ssc-miR-181b*	*CREB1*	3509	*3382*	−25.8
*ssc-miR-362*	*CTSB*	948	*718*	−25.8
*ssc-miR-185*	*LGMN*	1544	122	−25.7
*ssc-miR-185*	*SLA-DRB1*	341	*89*	−25.7
*ssc-miR-9813-5p*	*CD74*	2784	*1416*	−25.6
*ssc-miR-7*	*IL1B2*	583	*412*	−25.6
*ssc-miR-671-5p*	*SELL*	2974	*309*	−25.5
*ssc-miR-9813-5p*	*TNF*	1926	1378	−25.5
*ssc-miR-362*	*LGMN*	1544	*1280*	−25.4
*ssc-miR-9846-3p*	*CD80*	1492	*829*	−25.4
*ssc-miR-9846-3p*	*RFXANK*	1758	*1362*	−25.3
*ssc-miR-31*	*CD86*	3967	*1545*	−25.3
*ssc-miR-9816-3p*	*CD74*	2784	*757*	−25.3
*ssc-miR-9802-3p*	*SLA-DMB*	355	*52*	−25.2
*ssc-miR-99b*	*LGMN*	1544	*909*	−25.2
*ssc-miR-99b*	*CD74*	2784	*294*	−25.2
*ssc-miR-9802-3p*	*CTSB*	948	*39*	−25.0
*ssc-miR-181b*	*RFXANK*	1758	*308*	−25.0

**Table 2 ijms-21-00687-t002:** Primers used for miRNAs validation^1^_._

miR-9858-5p: forward (5′–3′) GCAGTTCCTGAGTCGGA	reverse (3′–5′) GGTCCAGTTTTTTTTTTTTTTTAGC
miR-671-5p: forward (5′–3′) CCCTGGAGGGGCTG	reverse (3′–5′) GGTCCAGTTTTTTTTTTTTTTTCCT
miR-30b-3p: forward (5′–3′) GCTGGGAGGTGGATGT	reverse (3′–5′) CAGGTCCAGTTTTTTTTTTTTTTTAAG
miR-181a: forward (5′–3′) CATTCAACGCTGTCGGT	reverse (3′–5′) GTCCAGTTTTTTTTTTTTTTTAACTCA
miR-30b-5p: forward (5′–3′) GCAGTGTAAACATCCTCGAC	reverse (3′–5′) TCCAGTTTTTTTTTTTTTTTCTTCCA

^1^ For primer design, the Busk [42] program was used.

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
