# Peer review of "The Probiotic BB12 Induces MicroRNAs Involved in Antigen Processing and Presentation in Porcine Monocyte-Derived Dendritic Cells"

_ijms, 2020, doi:10.3390/ijms21030687_

Round 1

Reviewer 1 Report

Bravo-Parra and colleagues present a study that has attempted to identify gene targets for miRNA whose expression is affected by stimulation with a probiotic BB12. The study may have merit; however, I have several concerns regarding the methodology of the work, which I believe needs to be addressed before this study gets published. My comments are listed below in the order of me reading the manuscript.

The introduction needs to be expanded and re-written. The authors should focus on the effect of probiotics (whether beneficial or not) on the immunity, and then focus on highlighting previous studies that have assessed miRNA expression in stimulated and non-stimulated immune cells. More importantly, by introducing such studies, the authors will lay the foundation for their current study, i.e., what is the hypothesis that the authors aim to test. The authors ought to provide a greater rationale for their reason to select the porcine model, vs. human or rodent MO? In my opinion, the sentence on lines 50-54, is insufficient to explain the choice of model. The two sentences on lines 54-56 are unclear. I'm not clear why the authors stimulated the monocytes with BB12 in the presence of TLR2 blocking. Were they trying to demonstrate that BB12 activates innate immunity through a mechanism different from TLR2? It is unclear to me whether the authors used the expression data as is or re-analyzed the GSE132995 data? If so, they need to explain how did they do that. Also, were the microarray expression data done for the protein-coding transcriptome as well. I am befuddled about the number of upregulated miRNAs! On line 77, the authors say they have used 33 up-regulated miRNAs, while on line 122 they talk about 44 up-regulated miRNA, which is? It is unclear to me whether the authors performed in silico analyses for all 33 up-regulated miRNAs? If so, how many targets were identified for all these miRNAs at their chosen mfe threshold? Considering that a singular miRNA can modulate the expression of hundreds genes, even at the chosen mfe the authors should have identified many gene targets for the 33 miRNAs. The authors report only 34 gene targets involved in MHC 2 pathway. What is the proportion of these targets among all gene targets identified? Further, while I understand the authors’ reason to focus only on the upregulated miRNA, identifying gene targets for the down-regulated miRNAs could also offer potential biological insight into the probiotic effects on immunity. My main concern with this study is that the main conclusions of the study are based only on in silico predictions. First, it has been recognized (for almost a decade now) that computational predictions have a very high false-positive rate when used alone. Ideally, the authors should have done experimental verification, if not for all gene targets, at least for their most significant findings. While that may not always be possible, at the least, they could have used gene and miRNA expression data from the same organism to perform a correlation-based analysis and intersect these with in silico based predictions to reinforce their findings.

While testing the expression of the miR-30b, -671, -9858 in BBL12 treated moDC is warranted (line 180), I think it is beside the point as the main premise of the paper is to demonstrate the ability of the selected miRNAs to interact with genes in immune-related processes. To that end it would have been much more compelling if the authors have attempted to validate these interactions empirically.

Reviewer 2 Report

An interesting study that reflects actual methods and research strategy. Well written introduction. Only a minor changes are needed in the methods and results - how was the specifity of amplicons in RT-qPCR of IL-10, PPIA and microRNAs checked and confirmed? Final conclusions are written only weak, rewritte it very clearly.

Author Response

Point 1: An interesting study that reflects actual methods and research strategy. Well written introduction. Only a minor changes are needed in the methods and results –how was the specifity of amplicons in RT-qPCR of IL-10, PPIA and microRNAs checked and confirmed?

Response 1: We thank the Reviewer for the positive comments. With respect to the specificity of amplicons, we verified them through the melt curves, looking for one peak and no amplification (no peaks) in the no template sample, to ensure the signal detected did not come from dimmers, formed during the hybridization of primers.

Point 2.Final conclusions are written only weak, rewritte it very clearly.

Response 2: The changes were done accordingly. Page 13, lines 434-445

Round 2

Reviewer 1 Report

I have no further comments to the authors